# EnCNN-UPMWS: Waste Classification by a CNN Ensemble Using the UPM Weighting Strategy

Hua Zheng [1] and Yu Gu [1,2,3,4,*]

1 College of Information Science and Technology, Beijing University of Chemical Technology, Beijing 100029, China; 2018200812@buct.edu.cn
2 Beijing Advanced Innovation Center for Soft Matter Science and Engineering, Beijing University of Chemical Technology, Beijing 100029, China
3 School of Automation, Guangdong University of Petrochemical Technology, Maoming 525000, China
4 Department of Chemistry, Institute of Inorganic and Analytical Chemistry, Goethe-University, Max-von-Laue-Str. 9, 60438 Frankfurt, Germany
* Correspondence: guyu@mail.buct.edu.cn

**Abstract:** The accurate and effective classification of household solid waste (HSW) is an indispensable component in the current procedure of waste disposal. In this paper, a novel ensemble learning model called EnCNN-UPMWS, which is based on convolutional neural networks (CNNs) and an unequal precision measurement weighting strategy (UPMWS), is proposed for the classification of HSW via waste images. First, three state-of-the-art CNNs, namely GoogLeNet, ResNet-50, and MobileNetV2, are used as ingredient classifiers to separately predict and obtain three predicted probability vectors, which are significant elements that affect the prediction performance by providing complementary information about the patterns to be classified. Then, the UPMWS is introduced to determine the weight coefficients of the ensemble models. The actual one-hot encoding labels of the validation set and the predicted probability vectors from the CNN ensemble are creatively used to calculate the weights for each classifier during the training phase, which can bring the aggregated prediction vector closer to the target label and improve the performance of the ensemble model. The proposed model was applied to two datasets, namely TrashNet (an open-access dataset) and FourTrash, which was constructed by collecting a total of 47,332 common HSW images containing four types of waste (wet waste, recyclables, harmful waste, and dry waste). The experimental results demonstrate the effectiveness of the proposed method in terms of its accuracy and *F1-scores*. Moreover, it was found that the UPMWS can simply and effectively enhance the performance of the ensemble learning model, and has potential applications in similar tasks of classification via ensemble learning.

**Keywords:** waste classification; ensemble learning; convolutional neural network; unequal precision measurement

## 1. Introduction

With the tremendous growth of the population and consumption, a huge amount of municipal solid waste (MSW) is generated every day, especially in developing countries [1]. MSW in developing countries is composed mainly of household garbage (55–80%) and commercial waste (10–30%) [2]. China has a population of 1.4 billion people and is the most populated developing country in the world. According to a survey conducted by the Ministry of Ecology and Environment of the People's Republic of China, the investigated 200 large and medium-sized cities generated 21,147.3 million tons of household solid waste (HSW) in 2018 [3]. Landfilling, the dominant waste disposal technology in China, has introduced serious water contamination to over half of the existing landfills due to the limited available land space in cities and the lack of high-cost permeate collection equipment in treatment systems [4]. As another important method by which to dispose of waste, incineration is expensive to operate and maintain, and also easily introduces

air pollution if there is a lack of air pollution control equipment [5]. Moreover, the large amount of dioxin emitted in the process of incineration aggravates global warming. Thus, solid waste disposal has become a challenging problem in China.

In the modern procedure of waste disposal, which includes waste separation, collection, transportation, and final treatment, recyclables and compostable waste account for 89.3% of HSW [6,7]. The accurate and efficient classification of waste can prevent waste pollution caused by mixing waste of different types, and can preclude the need for secondary sorting. As the initial point of the entire waste recycling process and the fundamental condition for ensuring effective recycling, the classification of waste can both enhance the efficiency of recycling and effectively protect the environment [8]. In other words, waste sorting is an effective way to reduce waste [9]. Because various types of waste require different types of disposal, a proper HSW standard is imperative in waste classification [10].

Waste classification is the procedure by which waste is assigned to specific classes based on its properties, characteristics, and/or components [6]. In past years, to effectively dispose of waste in China, the general criterion of municipal waste separation has been to divide waste into two types: recyclables and non-recyclables [6]. Nevertheless, to keep up with the practical demands of economic and environmental development, Beijing enacted a new waste classification policy in 2020 based on the policies of developed countries, including Japan [11] and Germany [12]. The new standard is to classify waste into four types, namely wet waste, recycling, harmful waste, and dry waste [13]. The ways by which to improve the efficiency of waste treatment rely not only on scientific criteria of waste classification, but also on the reliable and fast implementation of waste classification [14].

At present, waste classification mostly relies on inefficient manual work, which has many shortcomings, such as high work intensity, high cost, and potential harm to the health of workers [15]. Recently, automatic waste sorting and recycling facility systems based on the common sensor spectrum have been proposed. For example, Wu et al. [16] proposed an automatic plastic sorting system; they focused on sorting different plastics from waste electrical and electronic equipment based on near-infrared (NIR) spectroscopy. Additionally, Riba et al. [17] presented an approach for the sensing and classification of parts of an automatic waste textile sorting machine based on the infrared spectra of textile samples. These two studies respectively focused on the more refined classification of plastics and textiles. However, to gather spectral data for further processing, particular equipment is required, including NIR spectrometers, such as the NIR512 spectrometers by Ocean Optics, and the NIR radiation provided by a halogen light source. The equipment is expensive and complicated, and requires operation by professional personnel. In contrast, waste classification based on waste images via machine learning is accurate, simple, and convenient, and could therefore be used to construct an automatic smart waste sorter to alleviate the difficulties inherent in manual waste classification.

With the development of machine learning in recent years, deep learning has been widely used in speech recognition, visual object recognition, object detection, and many other fields [18–20]. A convolutional neural network (CNN) is a typical model that has been extensively used in image recognition and detection problems. Most recently, image recognition techniques in the computer vision field have been applied to waste classification. For instance, Xie et al. [21] proposed a framework based on a multilayer hybrid deep learning system (MHS) to recognize waste in urban public areas as recyclables or other types of waste. AlexNet [22] was used to extract representative features from waste images, and multiple functional sensors were used to obtain other information about the waste. While a high accuracy of over 90% was achieved, only two categories of waste were considered.

Thung et al. [23] released a dataset called TrashNet, which consists of 2527 images of waste divided into six different classes, namely glass, paper, plastic, metal, cardboard, and trash. The authors of Reference [24] proposed a model called RecycleNet, which achieved a classification accuracy of 81% on the TrashNet dataset. The authors of Reference [25]

proposed a combined model called Inception-ResNet, which achieved a classification accuracy of 88.6% on the TrashNet dataset. However, the size of this dataset is not large enough for deep learning, and easily leads to overfitting. Moreover, the performance results of these models have room for improvement.

In the context of classification tasks, ensemble-based methods have been employed to minimize the test errors [26]. There also exist several ensemble strategies for the combination of the prediction abilities of many different models. For example, Szegedy et al. [27] proposed an ensemble method that averages the softmax probabilities over all the individual classifiers to obtain the final prediction results, and this method was found to outperform single classifiers. Additionally, Chen et al. [28] adopted majority voting to combine models, and obtained a similar result. However, neither majority voting nor softmax probability averaging consider the differences between classifiers, and set the same weights for classifiers. Moreover, these methods may easily generate false predictions rather than correct predictions, as the integrated classifier regards the individual classifiers with the same reliability. In weight integration, each weight coefficient should be properly set, which is essential for the final prediction results [29].

Unequal precision measurement (UPM) is common in practice. Bar-Shalom et al. [30] proposed a one-step target tracking system solution for measurements obtained in discrete time, and Prieto et al. [31] proposed an adaptive likelihood method for robust data fusion in location systems. These models both fuse data by processing data of different types and with unequal precision, and parameter estimation can be improved with the assistance of data fusion. In the process of data fusion, the fusion weights of multiple heterogeneous unequal-precision data obtained under several different conditions are significant for the improvement of the precision of the measurement result [32].

Based on the preceding discussion, this paper proposes an ensemble learning model called EnCNN-UPMWS, which is based on three CNNs with different architectures and a UPM weighting strategy (UPMWS). The CNN ensemble couples the superior capabilities of the individual CNNs in terms of learning and exploring the patterns in waste image data, which improves the accuracy of the ensemble. Three state-of-the-art (SOTA) CNNs, namely GoogLeNet [27], ResNet-50 [33], and MobileNetV2 [34], are chosen as ingredient classifiers, and their performance on waste datasets is also demonstrated. To achieve further improvement in waste classification, the UPMWS, which involves the determination of the weights for UPM, is introduced in the CNN ensemble. It is worth mentioning that the UPMWS, which measures values in the process of data fusion, has never before been used in ensemble learning, let alone in an ensemble of CNNs. The main contributions of this work lie in the following three aspects:

1.  In this study, 47,332 images of waste belonging to four different classes, namely wet waste, recyclable waste, harmful waste, and dry waste, were collected from several open-access datasets and the Internet to create the FourTrash dataset;
2.  The proposed framework consists of several diverse SOTA CNNs (GoogLeNet, ResNet-50, and MobileNetV2) with different structures to deeply learn the features and explore the implicit information in waste images. These networks are treated as ingredient classifiers in the CNN ensemble;
3.  UPMWS is introduced to obtain reliable predictions by multiplying the result of each classifier and its corresponding weight coefficient. This can provide more robust results during the aggregation of the forecasting results of the CNNs.

The remainder of this article is organized as follows. Information about the materials and methods is provided in Section 2, and the proposed methods are presented in Section 3. The experimental results and discussion of this study are explained in Section 4. Finally, the conclusions are given in Section 5.

## 2. Materials and Methods

The related materials and methodologies employed in the proposed framework are introduced as follows.

## 2.1. Dataset

The performance of the proposed waste classification framework was evaluated on two datasets, namely the TrashNet dataset and the self-constructed FourTrash dataset. Each image in each dataset contains only one object. Hence, the aim of the two datasets is waste material classification, rather than the detection of waste items.

The TrashNet dataset was created by Thung et al. It contains 2527 images of waste divided into six different classes, namely glass, paper, plastic, metal, cardboard, and trash [23]. Some sample images are displayed in Figure 1.

The FourTrash dataset contains images of four different classes of waste, namely dry waste, wet waste, recyclables, and harmful waste. The 47,332 waste images in this dataset were partially collected from existing waste classification datasets [35], while other samples were collected from public websites. A few samples in this dataset are shown in Figure 2. Specifically, some objects of each class in the FourTrash dataset are described in Table 1.

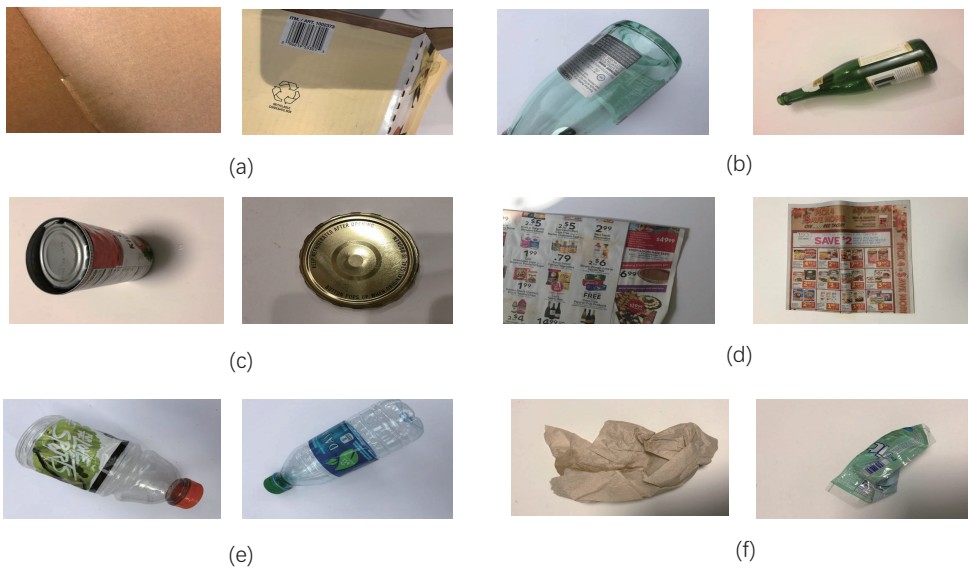

**Figure 1.** Images from the TrashNet dataset: (**a**) cardboard; (**b**) glass; (**c**) metal; (**d**) paper; (**e**) plastic; (**f**) trash.

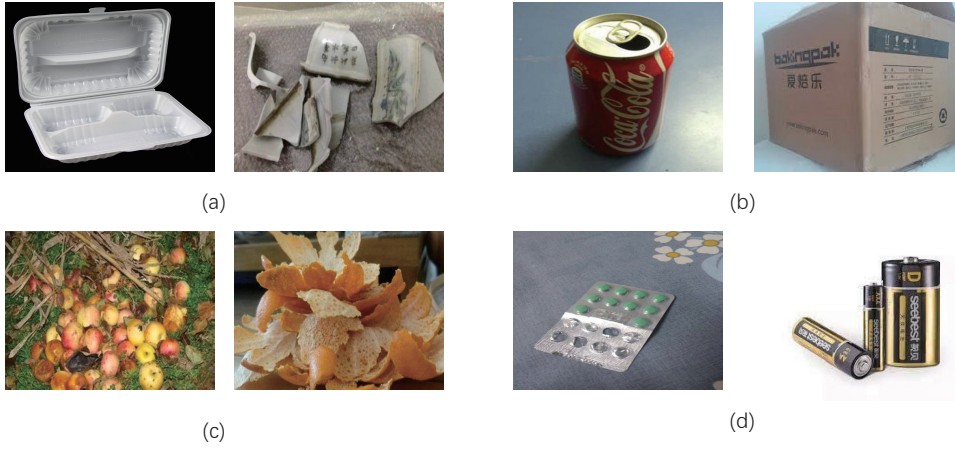

**Figure 2.** Images from the FourTrash dataset: (**a**) dry waste; (**b**) recyclables; (**c**) wet waste; (**d**) harmful waste.

**Table 1.** Some subclasses in the FourTrash dataset.

| No | Class | Objects |
|----|-------|---------|
| 1 | Recycling | power bank, bag, plastic toy, plastic basin, pop can, glass, carton, etc. |
| 2 | Dry waste | broken dishes, bamboo chopsticks, disposable fast food box, etc. |
| 3 | Wet waste | leftovers, fruit peel, vegetable leaves, eggshell, fishbone, etc. |
| 4 | Harmful waste | dry battery, ointment, expired drugs, glue, cosmetic packaging, etc. |

To validate the capacity of the proposed framework, the FourTrash dataset was randomly split into a training set (70%) and test set (30%). In the process of splitting, the dataset was divided based on subclasses, e.g., a bag of recyclables, instead of large categories, e.g., recyclables. The distribution results are shown in Table 2.

**Table 2.** Distribution of the FourTrash dataset.

| | Training | Test | Total |
|---|---|---|---|
| Recycling | 17,178 | 7361 | 24,539 |
| Dry waste | 3818 | 1636 | 5454 |
| Wet waste | 10,939 | 4692 | 15,631 |
| Harmful waste | 1196 | 512 | 1708 |
| Total | 33,131 | 14,201 | 47,332 |

*2.2. Convolutional Neural Network*

CNNs have been used as image classifiers in most computer vision fields that require a simple and high-accuracy classifier [36,37]. CNN models also serve as backbones in object detection [38–40].

In the 2012 ImageNet Large-Scale Visual Recognition Challenge (ILSVRC) competition, a deep CNN called AlexNet was proposed and achieved the best performance [22]. As a stunning result, the network achieved an error rate that was half the error rate of the best previous approach. This success brought about a revolution in computer vision. In recent years, many variant CNNs have been proposed in the ILSVRC competition, and the accuracy achieved by these models has almost reached its apex. In the following subsections, CNNs and three SOTA networks used in this work are respectively discussed.

2.2.1. A Brief Introduction of CNNs

Generally, a CNN architecture consists of convolutional layers, pooling layers, and fully connected (FC) layers [22]. Each convolutional layer extracts features from previous feature maps. Stacked convolutional layers are applied to extract feature maps from low-level abstraction to high-level abstraction [18]. The three types of layers and several important concepts in CNNs are subsequently introduced.

Convolutional layers are used to extract features from an input by applying convolutional operations. In these layers, convolution filters move over the feature map to generate features for the next layers, and the application of diverse convolution filters can yield different feature maps. The mathematical operation used in a convolutional layer can be expressed as follows:

$$X_i^l = \Sigma_{k=1}^{M_i} f(x_k^{l-1} * \omega_{ki}^l + b_i^l), \tag{1}$$

where $X_i^l$ represents the $i$-th feature map of the $l$-th layer, $x^{l-1}$ is the $k$ output feature maps of the former layer, and $\omega_{ki}^l$ represents the convolutional filter which used to map the $k$-th feature map in the $(l-1)$-th layer to the $i$th feature map in the next layer (the $l$-th layer). Additionally, the symbol "$*$" is the convolutional operator sign, $M_i$ denotes the size of the input, and $b_i^l$ denotes the bias of the convolutional layer. A nonlinear activation function, such as the rectified linear units (ReLU) function or sigmoid function, $f(\cdot)$, is commonly used in the convolutional layer.

The pooling layer is used after each convolutional layer, and it conducts sub-sampling to decrease the spatial size of the feature map and further minimize the number of parameters. The types of pooling include max-pooling and average-pooling operations. The max-pooling operation passes the maximum value in a local window, which can be defined as

$$P_i = \max_S X_i^l, \tag{2}$$

where $S$ is the size of the local window, and $X_i^l$ is the $i$-th feature map of the $l$-th layer. Via the operations mentioned previously, the CNN can achieve automatic feature extraction.

The FC layer connects the previous layer by flattening the features from the foregoing layer into a one-dimensional vector:

$$F^l = f(w^l(F^{l-1}) + b^l) \tag{3}$$

where $F^l$ represents the output of the $l$-th layer, $w^l$ represents the weight of the FC layer, and $b^l$ is the corresponding bias. Moreover, $f(\cdot)$ is a non-linear function.

At the top of the previous layers, a logistic regression function is used to construct a categorical output. The softmax layer connects the output of the previous layer, which is usually an FC layer, and generates a probability distribution of the categories via the softmax function as follows:

$$P_c(x) = \exp(y(x)) / \Sigma_{c=1}^C \exp(y(x)). \tag{4}$$

The operation in softmax layer can be expressed as

$$\mathbf{Z} = softmax(\mathbf{Y}), \tag{5}$$

where $\mathbf{Z}$ is the output vector that implies the probability of an element belong to the corresponding category, and $\mathbf{Y}$ represents the output of the last layer.

The cross-entropy loss function is usually used in classification tasks, and can be defined as

$$L = -\frac{1}{B} \Sigma_{i=1}^B \log(P(z = C_i | M_i), \theta), \tag{6}$$

where $B$ is the training batch size, $z$ is the output of the network, $(M_i, C_i)$ is a pair of input data and a label, and $\theta$ represents the weight parameters in the network that need to be updated. The CNN is trained by a certain number of iterations with gradient descent, and the weight parameters $\theta$ in the network can be updated until reaching the set threshold.

### 2.2.2. GoogLeNet

GoogLeNet [27] was proposed in the ImageNet Large-Scale Visual Recognition Challenge 2014 (ILSVRC14), and its architecture combines multiple convolutional layers with different filter sizes and pooling layers as a new module called the Inception module to increase the depth and width of the CNN. The main concept of Inception is to explore optimal local sparse structures in networks and determine how to cover them by the available dense components. As shown in Figure 3, an Inception module consists of $1 \times 1$ convolutions, which are used to reduce the model parameters and flexibly adjust the number of channels. Different filter sizes of $3 \times 3$ and $5 \times 5$ convolutions are designed to gain multi-scale features; thus, more discriminatory features can be obtained from the input data. These modules are stacked on each other, and their outputs tend to be diversified in terms of statistical correlation. One of the main advantages of this architecture is that the computational burden does not increase substantially despite the significant promotion of the representation capacity of the network. Another advantage is that the diverse sizes of filters can match various scales of visual information, and are then aggregated so that the next module can extract features from different scales.

GoogLeNet, the most successful instance, includes 9 stacked Inception modules, namely Inception (3a), Inception (3b), Inception (4a), Inception (4b), Inception (4c), In-

ception (4d), Inception (4e), Inception (5a), and Inception (5b). Additionally, several max-pooling layers are used between modules to adjust the dimensions of the outputs in the network. Eventually, the network is designed to be 22 layers deep, and can be efficiently run while consuming limited computational resources. Moreover, two auxiliary classifiers in the original GoogLeNet are connected to intermediate layers, and their loss is added to the total loss of the network. The operation of the two classifiers is an effective method by which to propagate gradients back through all the layers and obtain discriminatory features from relatively low-level layers.

Considering that the four-class classification task considered in the present study is less complex as compared to the classification tasks of the ILSVRC, which involves 1000-class objects, and to maintain the capacity of the network, only the output size of the last FC layer is changed to reflect the corresponding number of categories.

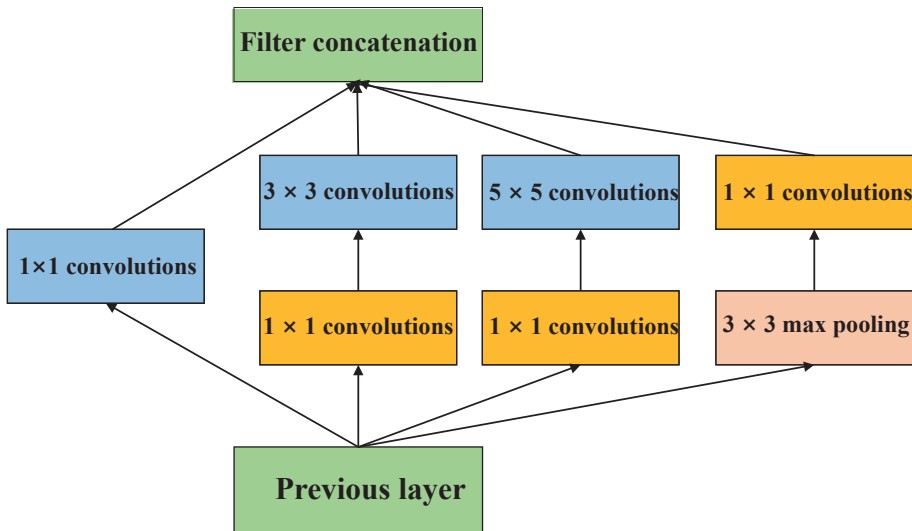

**Figure 3.** The Inception module.

### 2.2.3. ResNet

The residual neural network (ResNet) [33] is a network-in-network architecture that relies on many stacked residual blocks [41]. This module can alleviate the problem of the occurrence of a vanishing gradient when training very deep convolutional networks, and increases the relative depth of the network [41]. The main concept of this module is that it learns the difference between the input and output by adding a skip shortcut. In a plain CNN, the input of the $(l+1)$-th layer is generally used as the output of the $l$-th layer, which is denoted as $x_{l+1} = f(x_l)$. Unlike a plain CNN, ResNet adds a shortcut connection that performs identity mapping to the stacked layers, as shown in Figure 4. The output can be defined as

$$x_{l+1} = f(x_l) + x_l, \tag{7}$$

where $x_i$ is the input, and the function $f(\cdot)$ is the residual mapping to be learned. The residual module adds the input and the output, which has passed through some layers, together in one or $n$ steps. In this manner, ResNet stacks several residual blocks to obtain a deep CNN. Furthermore, the representation power of the network is tremendously promoted by increasing the depth, as well as by the residual blocks with shortcut connections.

ResNet-50 is a 50-layer version of ResNet, and uses 3-layer bottleneck containing $1 \times 1$, $3 \times 3$ and $1 \times 1$ convolutions, respectively. The $1 \times 1$ convolutional layer is used to adjust the dimensions. Similar to the modification of GoogLeNet, in this work, only the output dimension of the last FC layer is changed to the corresponding number of categories.

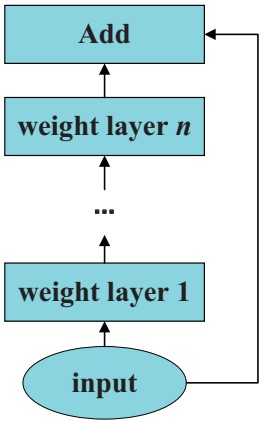

**Figure 4.** The residual block.

2.2.4. MobileNetV2

MobileNetV2 [34] is a lightweight model designed for mobile and embedded vision devices. The model is inherited from MobileNetV1 [42], and its main contribution is the introduction of a novel module, namely an inverted residual with a linear bottleneck.

The linear bottleneck removes non-linearities in the narrow layers, as non-linearities destroy information in low-dimensional space. Hence, the linear bottleneck maintains the power of representation. As shown in Figure 5, the input goes through a $1 \times 1$ convolution, depthwise $3 \times 3$ convolution, and $1 \times 1$ convolution, and a residual architecture is applied in the block. It should be noted that the first $1 \times 1$ convolution uses ReLU6 as a non-linearity instead of the ordinary ReLU, as ReLU6 is more robust when used with low-precision computation. Depthwise separable convolution [43] can be used to reduce the number of parameters, and thus makes computation effective. The inverted residuals in this architecture can relieve the vanishing gradient problem. The process of this module can also be described as a low-dimensional input first being expanded to a high dimension and filtered with a lightweight depthwise convolution. Features are then subsequently projected back to a low-dimensional representation with a linear convolution. From the theoretical aspect, this module also decouples the input/output domains from the expressiveness of the transformation [34].

MobileNetV2 stacks several bottlenecks and significantly decreases the number of operations and amount of memory. In this study, the last $1 \times 1$ convolutional layer in the original MobileNetV2 is substituted with an FC layer, the output dimension of which is the number of categories.

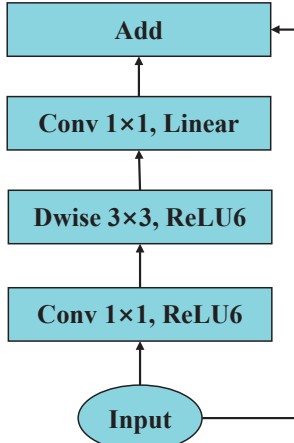

**Figure 5.** The convolutional block of MobileNetV2.

### 2.3. Multiple Classifier Integration

Multiple classifier integration originates from ensemble learning, and has been employed in some studies to achieve better generalization based on multiple base learners [44]. Furthermore, to obtain a better ensemble, the classifiers should be more diversified [45]. Two ensemble methods comprising bagging [46] (bootstrap aggregating) and boosting [47] are used to generate and exploit base learners. The concept of bagging was introduced in the work by Galar et al. [45]. It is based on bootstrap sampling to generate different training datasets from the original dataset. Consequently, different classifiers can be trained from these sampling sets. Regarding boosting, it has been demonstrated that weak learners can be promoted to strong learners by constantly adjusting the distribution of the training set [47]. During the process of training with adjustment, multiple base learners are generated. These classifiers are then combined via weighting.

Regarding classification tasks, multiple classifiers with different architectures potentially offer complementary information about the patterns to be classified [48]. Consequently, this could enhance the final performance of the integrated classifiers.

When combining several different base classifiers during the final period of ensemble learning, different combination strategies can yield different results. For example, bagging is a simple and comprehensive approach by which to obtain a strong learner. Its main concept is the construction of a strong classifier by combining multiple weak classifiers in a particular way [44]. Majority voting is usually used as a simple and effective combination method; each classifier first predicts a class based on a test sample, and the class with the most occurrences is then determined from these classes as the final prediction result. This method is simple and effectively benefits from integration [49]. The majority voting strategy generally directly exploits categorical labels. However, in the classification task, the predicted class possibility for sample $x$ can also be considered to be the output. For the $i$-th base learner, its prediction of one sample $x$ can be expressed as $P_i(x) : (p^{(1)}, p^{(2)}, \ldots, p^{(C)})$, where $p^{(n)}$ denotes the probability value for class $n$ and $C$ denotes the number of categories, $n \in (1, 2, \cdots, C)$. In this manner, weighted voting can be adopted as a combination strategy, as given by Equation (8), where $w_i$ is the weight coefficient of $P_i(x)$, $n$ represents the number of classes, and $T$ represents the number of classifiers.

$$H(x) = \arg \max_n \Sigma_{i=1}^{T} w_i P_i(x) \tag{8}$$

### 2.4. Weights in Unequal Precision Measurement

UPM is ubiquitous in practice. Generally, a series of measurements conducted under the same conditions is called an equal precision measurement. However, measurements are conducted under different conditions in most instances, e.g., by different personnel, with different instruments, and by employing different methods. Thus, the reliability of the measurement results will inevitably be different, and this type of measurement is called UPM.

The final result of equal precision measurement is an average of the measured values. However, UPMs usually have different reliabilities; thus, the average cannot be taken as the final result. To achieve more precise measurement results from observed UPM values, the weight coefficients of measurements under different conditions should be determined by uncertainty. Moreover, the uncertainty can be reflected by the variance of the measurement results.

If the different conditions in UPM are assumed to only reflect different measurement instruments, the weight coefficients of different measurement instruments can be determined by variances calculated by corresponding measured values for the same object. Theoretical results [32] indicate that the weights are inversely related to the variance of the measurement values, as given by Equation (9), where $w_j$ is the weight coefficient of the $j$-th instrument. In addition, the measured variance of $\sigma_j^2$ is calculated from a group of measured values. In this way, assuming that there are measurements $x_1, x_2 \ldots x_j$ respec-

tively measured by $j$ measuring tools, the final measurement result $X$ can be calculated by Equation (10).

$$w_1 : w_2 : \cdots : w_j = \frac{1}{\sigma_1^2} : \frac{1}{\sigma_2^2} : \cdots : \frac{1}{\sigma_j^2} \tag{9}$$

$$X = \Sigma_{i=1}^{j} w_i x_i \tag{10}$$

## 3. Proposed Method

This section describes the proposed waste classification model. First, in Section 3.1, the UMPWS is employed to obtain the weight coefficient of each classifier. Then, the main mechanism of the EnCNN-UPMWS model is described in Section 3.2.

### 3.1. UPMWS Method

The framework of the proposed CNN ensemble learning strategy is illustrated in Figure 6, and contains two stages. In the training stage, three classifiers with diverse architectures, namely GoogLeNet, ResNet-50, and MobileNetV2, are trained separately. As discussed previously, different classifiers can offer potential information concerning patterns to be classified. However, there exists no explicit theory for the use of diverse forecasting results from diverse classifiers. Most conventional ensemble models only employ simple averaging to integrate the prediction results. However, the forecasting performance of diverse classifiers can be improved if an appropriate method of weight setting is used. Furthermore, in the case of some samples in the four waste classes being similar and difficult to recognize, the three SOTA CNNs with powerful image classification capacity are used to recognize the type of waste.

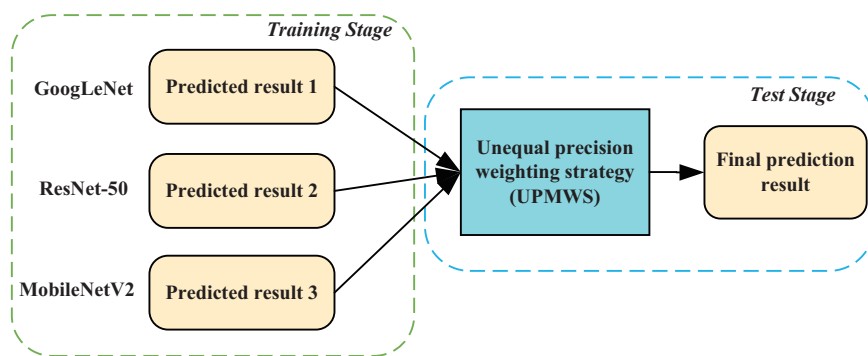

**Figure 6.** The structure of the ensemble learning strategy.

To make the most of each classifier and make the prediction results closer to the target, a method for setting the weights of UPM called UPMWS is proposed. According to the method described in Section 2.4, the UPMWS in the proposed ensemble strategy for the CNN classifiers is described as follows.

First, the input of the classifier is an image, and the output of each classifier is a probability vector. The dimension of the vector is the number of categories to be classified. After obtaining three probability vectors, weighted voting is employed as the ensemble strategy to aggregate these vectors to a final prediction vector in the testing phase. In general, each classifier is usually based on experience to set the weight. However, the differences between classifiers are not considered, and the performance of each classifier cannot be fully reflected. To a certain extent, if the classifier is treated as a measurement tool, the process of evaluating the classifier on all the samples in the validation set can be considered to be a measurement performed by a measurement tool.

Then, during the training process, a series of different prediction vectors on the validation set for each classifier is obtained. Furthermore, different classifiers often yield different prediction vectors. Therefore, these predictions can be regarded as classifiers with unequal precision obtaining different measurement results on the same validation

set. This process can be treated as UPM, which is a common problem in practice. In this way, for each classifier, the results of each valid metric should be considered to evaluate its reliability. In this article, the accuracy of the classifier in the validation set is used to determine whether its measurement is valid. In the training stage, each classifier is trained on the same training set and evaluated by the same validation set. Moreover, each classifier is evaluated once in each epoch. When the accuracy on the validation set is high enough, the prediction vectors and the one-hot encoding vector of the target label are used to calculate the weight coefficients. The UPMWS module presented in Figure 7 describes the proposed method by which to obtain the weight coefficients. The detailed calculation process is described as follows.

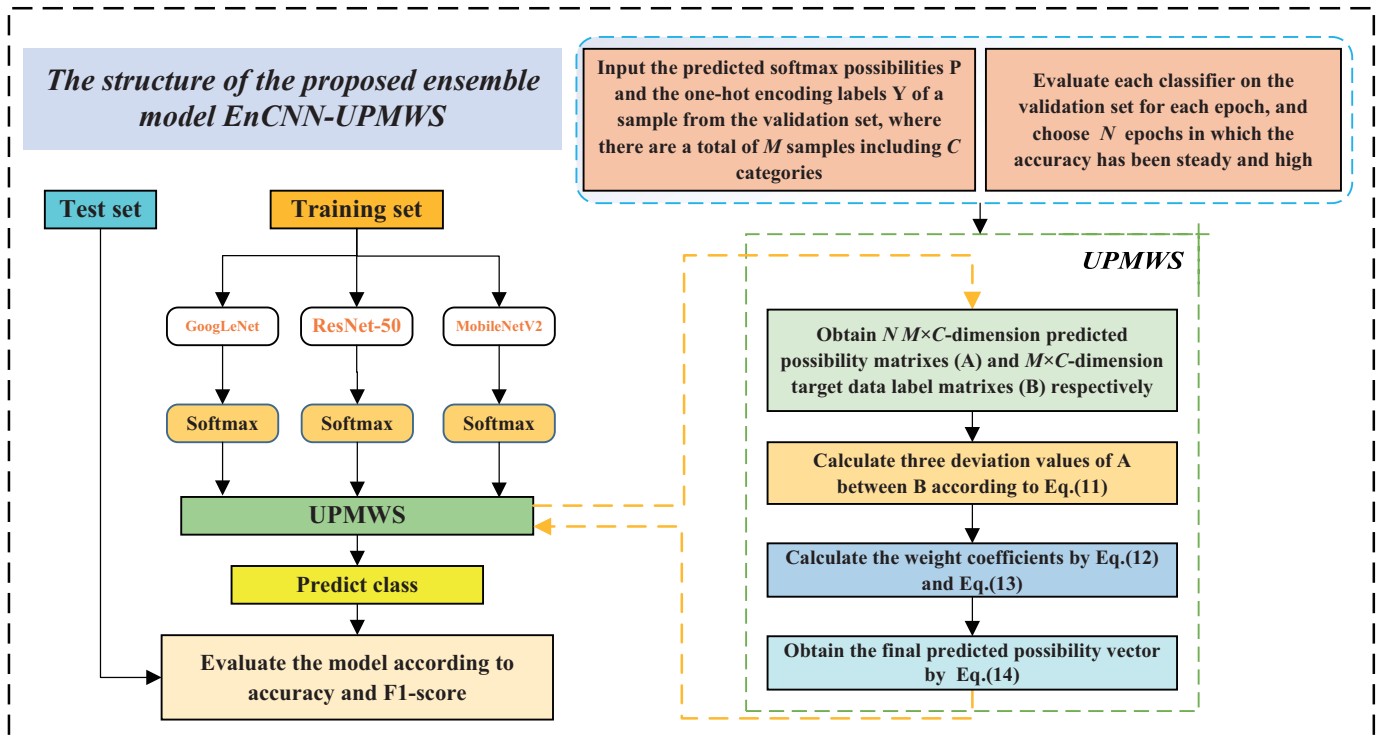

**Figure 7.** The structure of the proposed EnCNN-UPMWS.

For each model, the input is assumed to be $x$, the output is a C-dimension predicted probability vector **P**, and **Y** corresponds to a C-dimension one-hot encoding vector of the target label. Therefore, for all samples in the validation set, the measurement results can also be represented as $M \times C$ matrices (A), and the corresponding target data label can be denoted as $M \times C$ matrices (B). Moreover, a finite set $(1, 2, \cdots, C)$ is assumed, where $C$ denotes the number of classes. In this study, each model has $N$ results, the accuracies of which have reached a high level on the validation set during training. The actual label vector (one-hot) is viewed as the expected prediction value, and the deviation of the prediction results for each model is calculated as

$$\sigma = \frac{\Sigma_{n=1}^{N} \Sigma_{m=1}^{M} |\mathbf{Y}_m - \mathbf{P}_{(n,m)}|}{N}, \tag{11}$$

where $M$ is the number of samples in the validation set.

The weight coefficient $w_j$ of each model can be calculated by Equations (12) and (13) as follows.

$$\Sigma_{j=1}^{3} w_j = 1 \tag{12}$$

$$w_1 : w_2 : w_3 = \frac{1}{\sigma_1} : \frac{1}{\sigma_1} : \frac{1}{\sigma_3} \tag{13}$$

According to the obtained weight coefficients and the three predicted possibility vectors, the final prediction vector can be calculated as

$$Y^*(x) = \Sigma_{j=1}^3 w_j P_j(x), \tag{14}$$

where $x$ denotes the input data. Then, $Y^*$ is dealt with similar to Equation (8), and the final predicted label $P$ is calculated as

$$P = \arg\max_c Y^*(x), \tag{15}$$

where $c \in (1, 2, \cdots, C)$.

### 3.2. The Proposed Framework

This section presents an ensemble model based on three CNNs with diverse architectures and UPMWS integration for the improvement of the waste classification performance. The detailed structure of the proposed EnCNN-UPMWS model is shown in Figure 7. First, in the training stage, three SOTA CNN classification models, namely GoogLeNet, ResNet-50, and MobileNetV2, are trained on the training set. After sufficient training, three classifiers and corresponding weight coefficients are also obtained. The process of the training stage is presented in Figure 8.

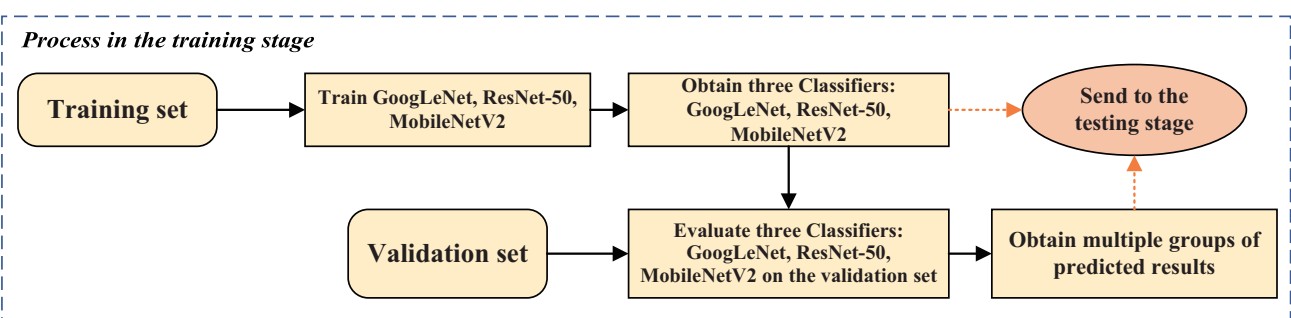

**Figure 8.** Process of the training stage in the EnCNN-UPMWS model.

In the testing stage, the three prediction results are interpreted by the UPMWS integration method, by which the weight coefficients are calculated and the final prediction results are obtained. The process of the testing stage is described in Figure 9.

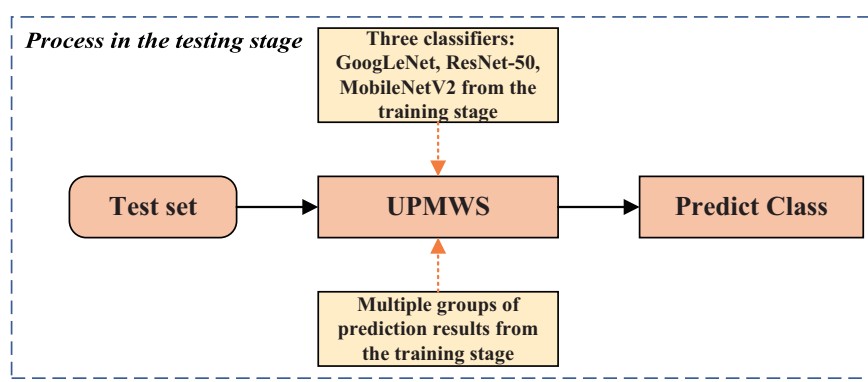

**Figure 9.** Process of the testing stage in the EnCNN-UPMWS model.

The structure of the proposed EnCNN-UPMWS model is shown in Figure 7, and its main steps of are described as follows:

1. Three diverse CNNs are used to explore the potential information concerning patterns of waste images to be classified. They are regarded as base learners in the CNN ensemble learning model;

2.   The UPMWS method, which can make the prediction result closer to the target, is introduced to obtain weight coefficient for each classifier. The details of UPMWS are given in Section 3.1;
3.   Three individual CNNs are used as competitors to demonstrate the classifying performance of the EnCNN-UPMWS by comparing it with GoogLeNet, ResNet50 and MobileNetV2 as well as the majority voting of the prediction results of the three CNNs.

## 4. Experimental Results and Discussion

### 4.1. Evaluation Metrics

In this study, the accuracy and two types of *F1-score* namely weighted and macro *F1-scores*, are used as performance metrics to assess the waste classification prediction performance. The accuracy is calculated by Equation (16):

$$Accuracy = \frac{(TP + TN)}{(TP + TN) + (FP + FN)}, \tag{16}$$

where *TP* (true positive) and *TN* (true negative) represent the numbers of samples correctly recognized as positive and negative, respectively, whereas *FP* (false positive) and *FN* (false negative) correspond to incorrectly estimated positive and negative samples, respectively. The *F1-score* can be interpreted as a weighted average of the precision and recall; its best value is 1, and its worst value is 0. The relative contributions of precision and recall to the *F1-score* are equal. The *F1-score* for each class is computed via the precision (*P*) and recall (*R*), which are defined as follows.

$$P = \frac{TP}{TP + FP} \tag{17}$$

$$R = \frac{TP}{TP + FN} \tag{18}$$

$$F1\ score = \frac{(2PR)}{(P + R)} \tag{19}$$

For multi-class cases, there are several ways to calculate the *F1-score* according to weighting, among which the macro and weighted F1-sores are chosen in this study. The macro *F1-score* is a metric calculated by the unweighted mean of each label, and is defined as

$$Macro\ F1\ score = \frac{\Sigma_{i=1}^{C} F1\ score_i}{C}, \tag{20}$$

where $F1\ score_i$ represents the *F1-score* of the *i*-th class, and C is the total number of classes. As shown in Equation (20), this metric is easily affected by a label with a high *F1-score*; thus, the weighted *F1-score* is introduced. The weighted *F1-score* takes label imbalance into account via support (the number of true instances for each label). The equation of this metric is as follows:

$$Weighted\ F1\ score = \Sigma_{i=1}^{C} w_i F1\ score_i, \tag{21}$$

where $F1\ score_i$ represents the prediction $F1\ score$ of the *i*-th class, and $w_i$ represents the weight coefficient, which is defined as:

$$w_i = \frac{N_i}{M}, \tag{22}$$

where $N_i$ is the total number of the *i*-th class for testing, and *M* represents the total number of samples for testing.

## 4.2. Results and Discussion

In this study, two sets of experiments, namely experiment A (FourTrash) and experiment B (TrashNet), were conducted on PyTorch, an open deep learning framework, to evaluate the performance of the proposed method. All coding was conducted on an NVIDIA GTX 2080Ti and Intel Xeon E5-2600 v4 3.60 GHz CPU. The input size of the images was resized to $224 \times 224$ to be compatible with GoogLeNet, MobileNetV2, and ResNet-50. To improve the convergence speed and generalization performance, the parameters of each model were initialized by a pre-trained model from ImageNet.

Furthermore, to promote the generalization performance of each classifier, some data augmentation strategies were respectively applied in the training process. These strategies included random center cropping, random rotation, and random horizontal and vertical flipping.

The same training configuration was used in the experiments, and is described as follows. The initial learning rate was set as $10^{-3}$, the Adam optimizer was used for training, and the cross-entropy loss function was used for the multi-class classification task in the training process. The number of a mini-batch was 64.

In both experiments A and B, the performance of the three CNNs with the corresponding datasets were first determined, and the results were compared with each other. The performance of the proposed method was then determined, and the results were compared with those of the majority voting method. Detailed performance information and comparisons with the proposed framework were reflected by the multiple metrics mentioned in Section 4.1.

### 4.2.1. Experiment A (FourTrash)

First, the original training set of the FourTrash dataset was split into a training set and validation set at a ratio of 9:1 to apply the UPMWS. To obtain a set of predicted probability vectors for each classifier, the accuracy trend of the validation set was observed to choose a proper range during training. The accuracy was found to achieve a high level from the 20th epoch, which indicates that the model had already been trained well, as shown in Figure 10. In this experiment, the epochs in the range of $[20, 50)$ were chosen to calculate the weight coefficients, and the results are reported in Table 3. The experiments reveal that GoogLeNet was more reliable than the other two models because its weight coefficient was the highest.

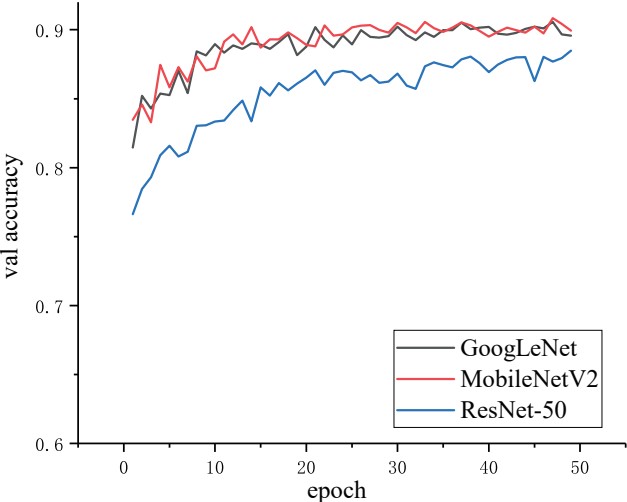

**Figure 10.** Validation accuracy during the training process on the FourTrash dataset.

The experimental results in terms of the accuracy and *F1-scores* are shown in Table 4. To analyze the performance of the UPMWS for the ensemble strategy, EnCNN-Voting

represents that each classifier voted for the final result based on the majority. Overall, the EnCNN-UPMWS framework outperformed the other approaches on the FourTrash dataset.

**Table 3.** Weight coefficients calculated by the UPMWS in Experiment A.

| Model | Weight Coefficient |
|---|---|
| GoogLeNet | 0.3872 |
| MobileNetV2 | 0.3588 |
| ResNet-50 | 0.2540 |

**Table 4.** Accuracy, macro and weighted *F1-scores* of different CNNs, EnCNN-Voting and EnCNN-UPMWS on the FourTrash dataset.

| Model | Macro *F1-Score* | Weighted *F1-Score* | Accuracy |
|---|---|---|---|
| EnCNN-UPMWS | **0.8825** | **0.9264** | **92.85%** |
| EnCNN-Voting | 0.8761 | 0.9216 | 92.32% |
| ResNet-50 | 0.8243 | 0.8878 | 89.00% |
| GoogLeNet | 0.8565 | 0.9081 | 90.97% |
| MobileNetV2 | 0.8417 | 0.8994 | 90.07% |

Specially, EnCNN-UPMWS achieved an accuracy of 92.85%, while ResNet-50, GoogLeNet, MobileNetV2, and EnCNN-Voting achieved accuracies of 89.00%, 90.97%, 90.07%, and 92.32%, respectively. Moreover, because the FourTrash dataset is extremely unbalanced, to evaluate the comprehensive performance of the models, the *F1-scores* of the models are also reported. Please note that the methods including the integration of CNNs achieved higher macro and weighted *F1-scores* than any single trained model. This demonstrates that both proposed ensemble methods can achieve better classification results by using the results of multiple classifiers. In addition, the macro and weighted *F1-scores* of EnCNN-UPMWS were 0.8825 and 0.9264, which were the best results among all the methods. This indicates that UPMWS is more effective than voting, which implies that giving the classifiers different weights can benefit more from the integration than simple averaging (i.e., setting the same weight).

To compare the classification results for each category, the *F1-score* for each waste category is reported in Table 5. Please note that EnCNN-UPMWS achieved the best *F1-score* in each class. To analyze the classification accuracy of each category, two confusion matrixes were obtained from the prediction results of voting and the proposed method, as presented in Figure 11. In a confusion matrix, the index of each column corresponds to a predicted label, and the indices of each row denote the actual label. Most incorrect predictions based on the proportion for the FourTrash dataset were made in the dry waste and harmful waste categories, which were misclassified as recyclable materials. The main reasons for the misclassifications include the following: (1) the numbers of samples of these two types were too few compared with recyclable and dry waste. As a result, the model could not learn enough information in these categories; (2) some subclasses in the two categories had similar characteristics with some subclasses in the recyclable category, i.e., the low inter-class differences concerning recyclable materials, harmful waste, and dry waste, which was also confirmed by the manual inspection of the related images.

**Table 5.** *F1-scores* of different CNNs, EnCNN-Voting, and EnCNN-UPMWS on the FourTrash dataset.

| Class | ResNet-50 | GoogLeNet | MobileNetV2 | EnCNN-Voting | EnCNN-UPMWS |
|---|---|---|---|---|---|
| Dry | 0.6775 | 0.7452 | 0.7391 | 0.7742 | **0.7845** |
| Wet | 0.9321 | 0.9441 | 0.9346 | 0.9529 | **0.9568** |
| Recyclable | 0.9142 | 0.9283 | 0.9215 | 0.9406 | **0.9442** |
| Harmful | 0.7734 | 0.8082 | 0.7714 | 0.8385 | **0.8445** |

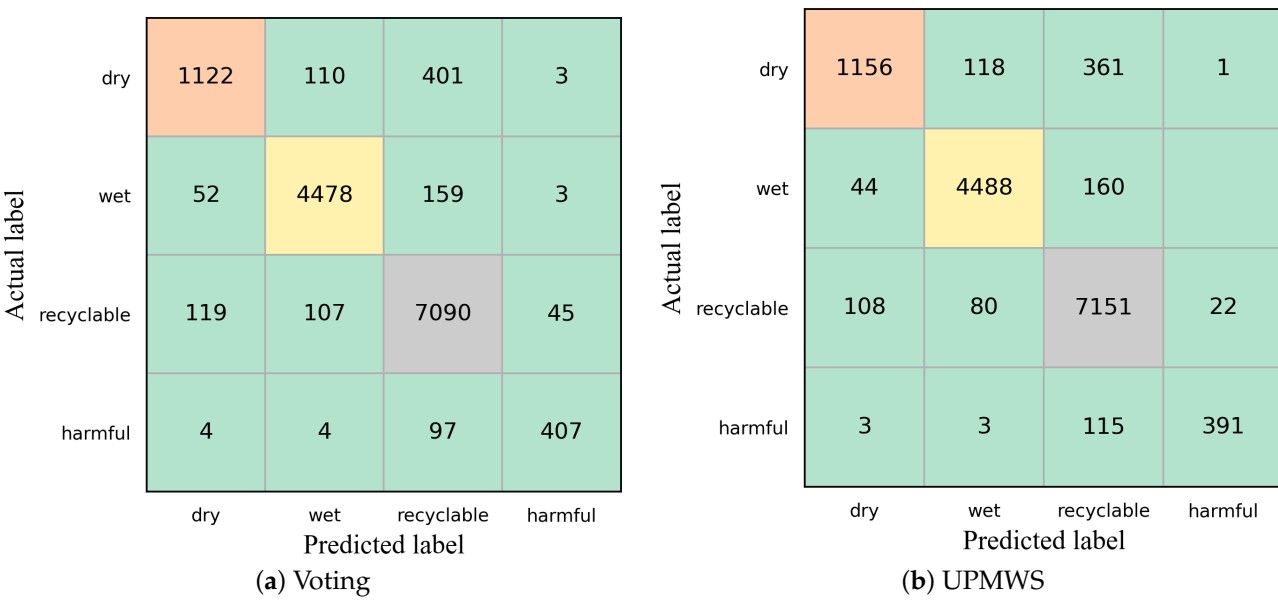

**Figure 11.** Confusion matrixes using (**a**) voting and (**b**) the UPMWS on the FourTrash dataset.

### 4.2.2. Experiment B (TrashNet)

TrashNet, an open-access waste image dataset, has been used to evaluate various image classification models [23]. First, to ensure the fairness of the comparison of the experimental results, the dataset splitting method described previously was used [25]. Similar to the process described in the previous section, to calculate the weight coefficients based on UPMWS, the trend of the accuracy of the validation set was observed. As exhibited in Figure 12, the accuracy reached a high level and then became steady. Therefore, epochs in the range of $[20, 50)$ were selected to calculate the weight coefficients, and the results are exhibited in Table 6. The results demonstrate that MobileNetV2 was more reliable than the other two models, as it had the highest weight coefficient. This also indicates that the same models exhibited different performance on different datasets.

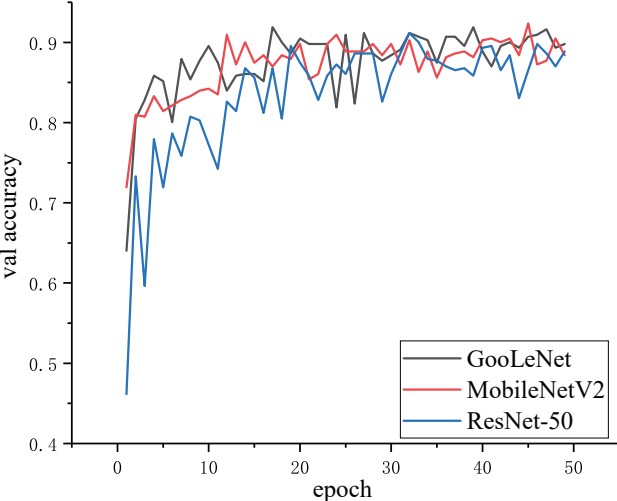

**Figure 12.** Validation accuracy during the training process on the TrashNet dataset.

**Table 6.** Weight coefficients calculated by the UPMWS in Experiment B.

| Model | Weight Coefficient |
|---|---|
| GoogLeNet | 0.3521 |
| MobileNetV2 | 0.3588 |
| ResNet-50 | 0.3011 |

As shown in Table 7, EnCNN-UPMWS exhibited good performance and achieved the best accuracy of 93.50%, which was higher than those of EnCNN-Voting (92.58%), ResNet-50 (90.95%), GoogLeNet (91.88%), and MobileNetV2 (91.42%). Its weighted *F1-score* was 0.9351, which was higher than those of EnCNN-Voting (0.9258), ResNet-50 (0.9099), GoogLeNet (0.9193), and MobileNetV2 (0.9136). Moreover, its macro *F1-score* was 0.9315, which was higher than those of EnCNN-Voting (0.9208), ResNet-50(0.8979), GoogLeNet (0.9125), and MobileNetV2 (0.9014). Furthermore, the classification results of the five models for each waste category (glass, paper, cardboard, plastic, and metal) were compared. As shown in Table 8, compared to the other models, EnCNN-UPMWS exhibited good performance in terms of the *F1-score*. The experimental results demonstrate the effectiveness of the UPMWS, which could improve the integration performance of CNNs.

Moreover, Figure 13 presents the confusion matrices of voting and the UPMWS. The UPMWS correctly classified more paper, cardboard, metal, and trash samples than did the voting strategy, whereas these methods performed the same for glass samples. Overall, EnCNN-UPMWS achieved the best performance on the TrashNet dataset.

**Table 7.** Comparative analysis of the classification indices of different CNNs, EnCNN-Voting, and EnCNN-UPMWS on the TrashNet dataset.

| Model | Macro F1-Score | Weighted F1-Score | Accuracy |
|---|---|---|---|
| EnCNN-UPMWS | **0.9315** | **0.9351** | **93.50%** |
| EnCNN-Voting | 0.9208 | 0.9258 | 92.58% |
| ResNet-50 | 0.8979 | 0.9099 | 90.95% |
| GoogLeNet | 0.9125 | 0.9193 | 91.88% |
| MobileNetV2 | 0.9014 | 0.9136 | 91.42% |

**Table 8.** *F1-scores* of different CNNs, EnCNN-Voting, and EnCNN-UPMWS on the TrashNet dataset.

| Indices | ResNet-50 | GoogLeNet | MobileNetV2 | EnCNN-Voting | EnCNN-UPMWS |
|---|---|---|---|---|---|
| Glass | 0.9036 | 0.9500 | 0.9125 | 0.9375 | 0.9317 |
| Paper | 0.9450 | 0.9252 | 0.9352 | 0.9401 | 0.9493 |
| Cardboard | 0.9481 | 0.9565 | 0.9429 | 0.9496 | 0.9565 |
| Plastic | 0.9116 | 0.8889 | 0.8974 | 0.9007 | 0.9139 |
| Metal | 0.8593 | 0.8921 | 0.9130 | 0.9078 | 0.9286 |
| Trash | 0.9197 | 0.8620 | 0.8077 | 0.8889 | 0.9091 |

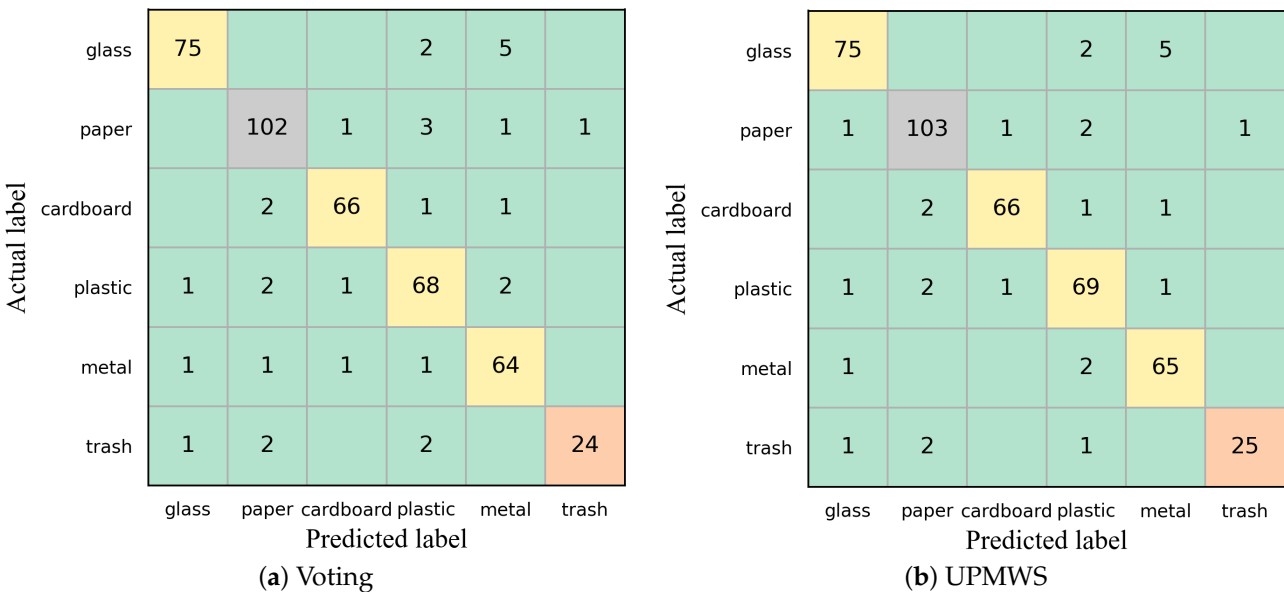

**Figure 13.** Confusion matrixes obtained using (**a**) voting and (**b**) the UPMWS on the TrashNet dataset.

The experimental results suggest that the integration of multiple classifiers can slightly promote the accuracy of waste image recognition. Multiple SOTA classifiers with diverse architectures can potentially offer complementary information about the patterns to be classified. The proposed UPMWS can be used to obtain more accurate measurement results, and different weight values are applied in combination with the prediction vectors; thus, a more accurate probability vector can be obtained. Ultimately, the overall framework was found to achieve good generalization and robust final prediction via the combination of the three classifiers.

## 5. Conclusions

In this paper, a framework (EnCNN-UPMWS) based on an ensemble learning strategy of three CNNs (GoogLeNet, ResNet-50, and MobileNetV2) and integration with the unequal precision measurement weighting strategy (UPMWS) was presented for HSW classification. In the proposed EnCNN-UPMWS model, three different types of CNN models are separately trained and saved. During training, the UPMWS is used to compute the weights for individual models. The three trained classifiers are then combined by adding the weighted predicted probability vectors together to obtain the final result for test samples. To evaluate the performance of the developed framework, it was compared with existing SOTA models in terms of four metrics (accuracy, *F1-score*, weighted *F1-score*, and macro *F1-score*) on two waste image datasets, namely FourTrash and TrashNet. In addition, the use of the majority voting method in the ensemble was also compared with the UPMWS.

Via the comparison of the results presented in Section 4, the proposed EnCNN-UPMWS was found to exhibit enhanced classification performance as compared to GoogLeNet, ResNet-50, and MobileNetV2. Moreover, the experimental results imply that the ensemble learning strategy outperformed the single models, and the proposed UPMWS method for weight setting outperformed the majority voting. On the FourTrash dataset, the overall accuracy of the proposed model for the four waste classes was 92.85%, which was 1.88% higher than the best accuracy of the single models and 0.53% higher than that of voting. Moreover, the macro and weighted *F1-scores* were respectively 0.8825 and 0.9264, which were respectively 0.026 and 0.0183 higher than the best indices of the single models and respectively 0.0064 and 0.0048 higher than voting. Furthermore, the proposed framework exhibited superior *F1-scores* for each class. For TrashNet, the overall accuracy of the proposed model for the six waste classes was 93.50%, which was 1.62% higher than the best accuracy of the single models and 0.92% higher than that of voting. Moreover, the

weighted and macro *F*1-*scores* were respectively 0.9351 and 0.9315, which were respectively 0.0158 and 0.019 higher than the best index of the single models and respectively 0.0093 and 0.0107 higher than voting. Furthermore, the proposed framework was superior to the other models in terms of the *F*1-*score* for most categories. Finally, the overall results demonstrate that the proposed EnCNN-UPMWS model can be considered to be a candidate for waste image classification.

The proposed UPMWS method, via which a set of proper weight coefficients is provided for base classifiers, works better than the majority voting method, via which the same weight coefficients are set for classifiers, and can therefore be applied in ensemble learning for classification tasks. In the future, the potential of the EnCNN-UPMWS model to solve more complicated tasks in waste image detection will be explored from the perspective of complex backgrounds.

**Author Contributions:** All authors contributed extensively to the study presented in this manuscript. H.Z. and Y.G. contributed significantly to the conception of the study. H.Z. designed the network and conducted the experiments. H.Z. and Y.G. provided, marked, and analyzed the experimental results. Y.G. supervised the work and contributed with valuable discussions and scientific advice. All authors contributed in writing this manuscript. All authors have read and agreed to the published version of the manuscript.

**Funding:** This research was funded by the Ministry of Science and Technology of the People´s Republic of China (Grant No. 2017YFB1400100) and the National Natural Science Foundation of China (Grant No. 61876059).

**Acknowledgments:** The authors would like to thank the Ministry of Science and Technology of the People´s Republic of China (Grant No. 2017YFB1400100) and the National Natural Science Foundation of China (Grant No. 61876059) for their support.

**Conflicts of Interest:** The authors declare no conflict of interest.

## Abbreviations

The following abbreviations are used in this manuscript:

| | |
|---|---|
| HSW | Household Solid Waste |
| CNN | Convolution Neural Network |
| UPMWS | Unequal Pecision Measurement Weighting Strategy |
| MSW | Municipal Solid Waste |
| ResNet | Residual Neural Network |
| MHS | Multilayer Hybrid deep learning System |
| NIR | Near-infrared |
| ReLU | Rectified Linear Units |
| FC | Full Connected |
| SOTA | State-Of-The-Art |
| ILSVRC14 | ImageNet Large-Scale Visual Recognition Challenge 2014 |
| bagging | bootstrap aggregating |
| EnCNN-UPWMS | a CNN Ensemble using the UPM Weighting Strategy |
| EnCNN-Voting | a CNN Ensemble using the majority Voting method |
| P | Precision |
| R | Recall |
| TP | True Positive |
| TN | True Negative |
| FP | False Positive |
| FN | False Negative |

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
