# Peer review of "EnCNN-UPMWS: Waste Classification by a CNN Ensemble Using the UPM Weighting Strategy"

_electronics, doi:10.3390/electronics10040427_

Round 1
Reviewer 1 Report
Comments:
- In keywords, replace abbreviation CNN with the full name;
- Explain all abbreviations in the article;
- Improve the quality of all diagrams from 3 to 9 (bold or enlarge the descriptions in diagrams, please;
- Improve also the quality/clarity of the subsequent figures;
- How the results obtained on the basis of the proposed method are assessed in comparison to the literature data? Present a brief discussion in section 4.2.2, please.
- In summary, answer the question: what are the chances of the UPMWS application?
Reviewer 2 Report
An interesting and relevant field of work is addressed. The work on classifier fusion in the context of CNN in general is interesting.
The choice of data from visual range only and rather simple single object/background scenes seem to be quite special. Are there references on the real-world scenario such a system would have to meet ?
A clear goal definition would be salient. How would the proposed system react to objects arriving with occlusion on a conveyor belt ? What would happen on the presentation of object categories not present in the learning set and/or objects torn to pieces ?
References to the common sensor spectrum applied in waste sorting and recycling facilities would be improving the paper.
Round 2
Reviewer 2 Report
An interesting and relevant field of work is addressed. The work on classifier fusion in the context of CNN in general is interesting.
How would the proposed system react to objects arriving with occlusion on a conveyor belt ? What would happen on the presentation of object categories not present in the learning set and/or objects torn to pieces ?
